# Mesenchymal Stem Cells and Secretome as a New Possible Approach to Treat Cartilage Damage: An In Vitro Study

**DOI:** 10.3390/biom14091068

**Published:** 2024-08-26

**Authors:** Valentina Bina, Alice Maria Brancato, Laura Caliogna, Micaela Berni, Giulia Gastaldi, Mario Mosconi, Gianluigi Pasta, Federico Alberto Grassi, Eugenio Jannelli

**Affiliations:** 1Department of Molecular Medicine, University of Pavia, 27100 Pavia, Italy; valentina.bina01@universitadipavia.it (V.B.); giulia.gastaldi@unipv.it (G.G.); 2Orthopedics and Traumatology Clinic, IRCCS Policlinico San Matteo Foundation, 27100 Pavia, Italy; alicemaria.brancato01@universitadipavia.it (A.M.B.); mario.mosconi@unipv.it (M.M.); gianluigipasta@yahoo.it (G.P.); federico.grassi@unipv.it (F.A.G.); e.jannelli@smatteo.pv.it (E.J.); 3Department of Clinical, Surgical, Diagnostic and Pediatric Sciences, University of Pavia, 27100 Pavia, Italy; micaela.berni@hotmail.com; 4Centre for Health Technologies, University of Pavia, 27100 Pavia, Italy

**Keywords:** chondrocyte, secretome, SVF, cartilage repair, ECM proteins

## Abstract

**Introduction**: Osteoarthritis is a degenerative condition of the cartilage, often common among the population and occurs frequently with aging. Many factors are decisive for the development of its pathogenesis such as age, obesity, trauma, mechanical load, and modification of synovial biology. The main features of osteoarthritis are chondrocytes and cartilage matrix loss, which lead to pain, loss of function of the whole joint, and disability, representing a relevant health problem. Recently, a new therapeutic approach based on cell therapy has been studying the regenerative ability of mesenchymal stem cells for osteoarthritic chondrocytes. **Aim**: This in vitro study clarifies the regenerative effects of multipotent adipose-derived stem cells and the pluripotent amniotic epithelial stem cells on arthrosis chondrocytes by performing co-culture experiments. **Methods**: We studied the regenerative potential of secretome (soluble factors and extracellular vesicles), mesenchymal stem cells, and the adipose stromal vascular fraction. The regenerative effects were evaluated by gene and protein expression analysis of articular cartilage-specific genes and proteins like *col2a1*, *acan*, and *sox9*. **Results**: Mesenchymal stem cells, secretome, and adipose stromal vascular fractions influenced the cartilage genes and protein expression. **Conclusions**: The results indicate that the treatment with mesenchymal stem cells could be the best biological approach for cartilage regenerative medicine.

## 1. Introduction

Cartilage is an avascular, supporting, and articular component of the musculoskeletal system, characterized by a dense extracellular matrix (ECM) and chondrocytes. It has a key role in the biochemical network, regulating a variety of processes from morphogenesis to tissue homeostasis. The ECM generally contains two classes of proteins: proteoglycans (PGs) and fibrous proteins (the main ones are collagen, fibronectin, elastin, and laminins) [1,2,3].

Articular cartilage (AC) is the highly specialized connective tissue found in the diarthrodial joints. Its primary function is to provide a smooth and lubricated surface between the bones to distribute the loads on a surface with a low frictional coefficient.

The composition of the ECM and the response of chondrocytes to external factors change with aging. In the elderly, cartilage chondrocytes accumulate in the deeper regions, and the ECM contains smaller proteoglycan aggregates with a decreased amount of water with consequent increased stiffness and impaired resistance to compressive forces [4].

While articular cartilage contains many progenitor cells, it has shallow regenerative capabilities with a consequent susceptibility to malfunctions after acute injury or chronic inflammation [5,6]. The articular cartilage consists of resting chondrocytes characterized by a low proliferation rate and by the expression of articular cartilage-specific genes, such as *col2a1*, *acan*, and *sox9*.

Osteoarthritis (OA) is the most prevalent degenerative condition of the cartilage that affects more than 300 million people. Age, obesity, trauma, mechanical load, and modification of synovial biology are relevant to its pathogenesis. The main features of osteoarthritis are cartilage loss, osteophyte formation, subchondral bone alterations, and synovial inflammation. OA is a condition characterized by chondrocyte hypertrophy and cell death due to stress signals, such as inflammation, injuries, infections, and obesity, which leads to progressive joint disruption. Angiogenesis across the osteochondral junction may contribute to articular cartilage changes as it appears to be associated with the loss and depletion of proteoglycans, as well as the deposition of collagen types I and X, responsible for cartilage calcification or ossification.

The formation of new vessels is also involved in the modulation of chondrocyte activity and in the degradation of the cartilage matrix [7]. Cytokines and chemokines play an important role in OA as they are responsible for cartilage damage and inflammation of the synovia [8]. The main ones are IL-1, involved in the destruction of cartilage through the inhibition of the synthesis of aggrecan and collagen and the promotion of proteolytic enzymes; y-chain cytokines, responsible for the recruitment and survival of lymphocytes; IL-15, associated with the progression of the disease; and IL-7, relevant in the production of matrix metalloproteinases. The main chemokines involved in OA are IL-8, CCL5, and CCL19, responsible for the recruitment and trafficking of inflammatory cells [9,10]. Other potentially relevant factors in this pathology are mitochondrial dysfunction, molecular patterns associated with damage, metabolites, and crystals [11].

In the early stages of OA, the chondrocytes start expressing runx2, col10a1, and a series of matrix metalloproteases (MMPs). MMPs are involved in the breakdown of the extracellular matrix. When damage starts, the resident chondrocytes partially de-differentiate to try to repair small lesions, increasing the synthesis of collagens and proteoglycans and acquiring a fibroblast-like phenotype. If the lesion is too large, the chondrocytes become proliferative and undergo hypertrophic differentiation, characterized by cell death or osteogenic differentiation, ECM mineralization, vascular invasion, and ECM remodeling [12,13].

Actually, conventional OA therapies consist mainly of symptom management, especially pain, rather than promoting the regeneration of the damaged cartilage [14]. The most widely used pharmacological therapies in osteoarthritis symptom management are the nonsteroidal anti-inflammatory class (NSAIDs), corticosteroids, and visco-supplementation treatments with hyaluronic acid. For patients who have not responded to any therapy, surgery is recommended.

Recently, a new therapeutic approach based on cell therapy is emerging.

New insights studies show that mesenchymal stem cells and pluripotent stem cells have regenerative capabilities for cartilage damage [15,16,17].

The aim of this article was to elucidate the molecular mechanisms underlying the pro-degenerative and pro-regenerative processes executed by the treated chondrocytes, exploring the contribution of the stem cells. We focused our attention on two extremely interesting cell lines: multipotent adipose-derived stem cells (hASCs) and pluripotent amniotic epithelial stem cells (AESCs). This choice has been dictated by the clinical demands; indeed, both cell lines have minimal ethical issues, low immunological rejection rates, and ease of isolation compared to other cell populations.

We performed co-culture experiments using both hASCs and AESCs co-cultured with diseased chondrocytes (OA) to test the possible regenerative effects of these stem cells. Moreover, we studied the regenerative potential of the hASC’s secretome (both soluble factors and extracellular vesicles) and the adipose stromal vascular fraction (SVF), containing several progenitor cells and blood-derived components. The regenerative outcomes were evaluated using gene and protein expression analysis and cartilage regeneration was studied using healthy, fibrotic, and hypertrophic chondrocyte markers.

## 2. Materials and Methods

Different types of cells were used in this study:Healthy chondrocytes purchased from CliniSciences (CliniSciences S.r.l. Guidonia Montecelio, Italy);Osteoarthritic chondrocytes isolated from patients;Amniotic epithelial stem cells (AESCs) isolated from the amniotic membrane of healthy donors, kindly donated by obstetrics unity of Poliambulanza Foundation, Brescia;Human adipose-derived stem cells (hASCs) isolated from patients.

Isolation of osteoarthritic (OA) chondrocytes. OA chondrocytes were isolated from biopsies of articular cartilage (AC) during total knee replacement surgery. Informed consent was obtained from all patients before surgery. The study was conducted according to the 1975 Declaration of Helsinki and approved by the Ethics Committee of the San Matteo Foundation, Research and Care Institute, Pavia, Italy (P-20170004278, 1 March 2017).

The AC was washed twice with sterile PBS 1X and cut with surgical scissors. To digest the resident chondroblasts, the cartilaginous slices were incubated for 2 h in a shaking bath (140 rpm) at 37 °C with 0.025% trypsin. After the incubation, the AC was washed several times with sterile PBS 1X to eliminate the trypsin residues. Therefore, the cartilage was incubated with 0.02% collagenase II Clostridium histolyticum in High Glucose DMEM supplemented with 100 U/mL of penicillin, 100 g/mL of streptomycin, and 0.25 g/mL amphotericin B for 20 h in a shaking bath at 140 rpm and 37 °C. At the end of the digestion, the cellular suspension was filtered with a 100 μm membrane pore size and then washed with High Glucose DMEM and centrifuged twice (1200 rpm, 10 min at 20 °C). Finally, the pellet containing chondrocytes was suspended in High Glucose DMEM supplemented with 100 U/mL of penicillin, 100 g/mL of streptomycin, 0.25 g/mL of amphotericin B, and ascorbic acid (0.05 mM).

Isolation of human adipose-derived stem cells (hASCs). Subcutaneous adipose tissue was obtained from the peri-trochanteric region of healthy donors during hip replacement surgery. Informed consent was obtained from all patients before surgery (Ethics Committee number P-20170004278, 1 March 2017). The sample was washed twice with sterile PBS 1X, finely minced with surgical equipment, and incubated with 0.02% collagenase II Clostridium histolyticum in DMEM-F12 HAM supplemented with 100 U/mL of penicillin, 100 g/mL of streptomycin, 0.25 g/mL and amphotericin B for 1 h in a shaking bath at 140 rpm and 37 °C. After the incubation and neutralization with DMEM F-12 HAM, the cell suspension was filtered with a 100 μm pore size membrane to remove the non-digested debris and then centrifuged twice at 1200 rpm for 10 min at 20 °C. At the end of the washing steps, the pellet was treated with a lysis solution for 10 min, suspended in DMEM F-12 to neutralize the lysing effect, and centrifuged for 10 min at 1200 rpm at 4 °C. The obtained pellet was the stromal vascular fraction (SVF) that also contained hASCs. Half of this pellet was used in the co-culture experiment, and the other half was suspended in growth medium (GM) DMEM F-12 HAM supplemented with pen/strep and amphotericin B, seeded in flasks and cultured in a humidified atmosphere with 5% CO_2_ at 37 °C to obtain hASCs.

Healthy chondrocytes. Human healthy chondrocytes purchased from CliniSciences were kept in a chondrocyte growth medium (CliniSciences M2600-2HS) for up to 48 h before seeding the 3D micromasses.

Secretome. The secretome from hASCs was purchased from Pharma Exceed Translational Research.

Cell seeding and culture. Three-dimensional culturing systems (micromasses) were prepared to evaluate the effects of stem cells (SCs) on OA chondrocytes while helping the primary cells retain their phenotype as much as possible. The 100% SC and 100% OA chondrocyte pellets were used as inner controls, while SCs (both hASCs and AESCs) and chondrocytes were co-cultured in a 1:1 ratio. Each micromass contained ca. 100,000 cells per cell line. The cells used in this experiment were in passage 2. The pellets were cultured at 37 °C in a humidified atmosphere (5% CO_2_) in conic tubes, completely soaked by the GM (DMEM high glucose, 10% FBS, pen/strep, and amphotericin B) supplemented with ascorbic acid (0.05 mM). The ascorbic acid was added daily to maintain its optimal concentration and the GM was refreshed once a week. For each condition (OA with hASCs, OA with AESCs, OA with SVFs, and OA with secretomes) we performed 6 replications.

After 7 and 21 days of culture, the micromasses were sacrificed to extract the total RNA, while the micromasses were sacrificed after 21 days for the whole protein content.

RNA isolation and reverse transcriptase quantitative real-time PCR (qRT-PCR). The gene expression analysis was performed using RT-PCR, and the genes investigated were the following: collagen type I (*col1a1*), collagen type II (*col2a1*), collagen type X (*col10a1*), aggrecan (*acan*), the transcription factors SOX9 (*sox9*) and RUNX2 (*runx2*), the matrix proteases MMP1 (*mmp1*) and MMP13 (*mmp13*), and the housekeeping gene beta-2-microglobulin (*B2M*). The total RNA was extracted from the micromass pellets after 7 and 21 days of in vitro culture using TRI-Reagent according to the Chomczynski method. The RNA was reverse-transcribed into cDNA using random hexamers and M-MLV Reverse Transcriptase. Quantitative real-time PCR (qRT-PCR) was performed in triplicate using 2 μL of cDNA, obtained as above, using specific primers from Qiagen (Qiagen, Hilden, Ger-many) reported in Table 1. The cycling conditions that we used to perform the qRT-PCR were the following: denaturation at 95 °C for 5 min; denaturation at 95 °C for 30 s for 40 cycles; 60 °C for 30 s for the annealing; and 72 °C for 40 s for the elongation. The results of the qPCR were normalized against the expression of the housekeeping gene (B2M) to obtain the relative gene expression, reported as ΔCt. Then, the values were compared against the internal biological control to obtain the ΔΔCt. Finally, a formula (2^−ΔΔCt^) was used to obtain the fold change (FC), which is a value directly proportional to the relative gene expression.

ELISA assay. The secretion of structural proteins was assessed by testing the relative concentration of ECM proteins in the cultured micromasses. Protein synthesis and secretion were quantified using ELISA assay kits purchased from Cloud-Clone Corp., Katy, TX, USA. At the end of the in vitro culture (21 days), the pellets were centrifuged, and the supernatant was discarded. The micromasses were stored at −20 °C until their further processing. The freezing step helped to crack the micromass favoring protein release and avoiding the use of a lysis buffer that might have interfered with the ELISA assays. All the samples were tested undiluted, apart from those destined for the COL1A1 assay (dilution 1:10). The downstream processing was in accordance with the manufacturer’s instructions. Briefly, the samples were incubated on the primed wells and then treated with a biotin-conjugated primary antibody. Next, to detect the signal arising from each well, avidin conjugated to Horseradish Peroxidase (HRP) was added to each well, incubated, and an appropriate substrate was added. The intensity of the developed color was read with a spectrophotometer with a λ of 450 nm ± 10 nm. The absolute concentration was then normalized against total protein content, measured via Coomassie Brilliant Blue staining. The purchased kits were targeting collagen type I (SEA571Hu 96 Tests), osteopontin (SEA899Hu 96 tests), osteocalcin (SEA471Hu 96 tests), osteonectin (SEA791 Hu), cartilage oligomeric protein (SEB197Hu), collagen type II (SEA572Hu), and collagen type X (SEC156Hu).

Statistical Analysis. All the statistical analyses were performed with GraphPad Prism 8.4.2. All the data were tested for normality and subsequently analyzed with unpaired *t*-tests, a one-way ANOVA followed by Tukey’s and Dunnett’s tests for multiple comparisons, or by simple linear regression. The differences were considered statistically significant when *p* < 0.05.

## 3. Results

### 3.1. Gene Expression Analysis

#### 3.1.1. Co-Culture between OA and hASCs and Co-Culture between OA and AESCs

QRT-PCR analysis was conducted for three different conditions: co-culture between hASCs and OA, co-culture between AESCs and OA, and as a control, OA chondrocytes, after 7 days post-seeding (dps) and after 21 days post-seeding (the end of the culturing time). The results of the co-cultures are reported as FCs (Fold changes), calculated over the mean of gene expression values of the single cell line.

The healthy chondrocyte markers, such as *sox9*, *col2a1*, and *can*, were evaluated at 7 and 21 dps. OA chondrocytes with hASCs showed an increase in *sox9* expression at 21 dps and a boost in *acan* expression at 7 and 21 dps. Meanwhile, OA chondrocytes co-cultured with AESCs showed an increase in *acan* and *col2a1* expression at 7 dps (Figure 1).

The degenerative outcome of OA was evaluated by testing two well-known hypertrophic markers: *runx2* and *col10a1*. The analysis showed a significant downregulation *runx2* in the OA + hASCs and an upregulation in OA + AESCs at 7 and 21 dps. In addition, co-culture with AESCs showed a strong increase in *col10a1* expression at 21 dps (Figure 2).

The gene expression of *col1a1*, *mmp1*, and *mmp13* were analyzed to investigate the fibrotic response. The OA chondrocytes co-cultured with the AESCs displayed a high expression of *col1a1* both at 7 and 21 dps. Also, there was an earlier expression of *mmp1* (7 dps) and a later expression of *mmp13* (21 dps), while hASCs induced an increased expression of *mmp13* at 7 dps, which was downregulated at 21 dps (Figure 3).

#### 3.1.2. Co-Culture between OA and Hascs, Co-Culture between OA and Svfs, and Co-Culture between OA and Secretome

To better clarify the molecular mechanisms responsible for tissue regeneration, the regenerative capacity of the secretome and the stromal vascular fraction (SVF) were investigated. The gene expression of the co-culture with OA and hASCs, OA with SVFs, and OA with the secretome were compared. The results indicated that there were no meaningful differences in the expression of the healthy-cartilage markers (*sox9*, *acan*, and *col2a1*) of chondrocytes cultured in the presence of hASCs or with the secretome or SVF compared to the untreated cells (Figure 4).

The treatment with the SVF induced a fibrotic condition because, in co-culture with OA, there was a significant increase in *col1a1* expression along with no changes in collagenase activities (Figure 5).

The SVF also had a role in degenerative markers because it induced an increase in *col10a1* expression both at 7 dps and 21 dps and an increase in *runx2* at 21 dps (Figure 6).

### 3.2. ELISA Assay

The proteins secreted by chondrocytes were studied to understand the molecular pathways and metabolic activities involved with healthy or osteoarthritic chondrocytes. At 21 dps, we performed an ELISA quantitative analysis of the ECM proteins and cartilage-specific proteins COL1A1, COL2A1, COL10A1, and COMP, secreted by both healthy and OA chondrocytes. The results were normalized against the total protein content. The Coomassie Brilliant Blue staining was used to quantify the total protein. The pellets were resuspended in 1 mL of Coomassie and read using a spectrophotometer at 595 nm. The relative protein concentration was expressed as a percentage of the proteins secreted by healthy chondrocytes. The results showed a significant statistical increase in the total amount of proteins secreted by OA chondrocytes with respect to healthy ones (Figure 7A). At the same time, it was evident that there was a decrease in OA ECM proteins despite the healthy chondrocytes (Figure 7B). The results showed a higher level of all proteins tested (COL1A1, COL2A1, COL10A1, and COMP) in healthy chondrocytes, with results statistically significant only for COL2A1 and COMP (Figure 7C).

To investigate the role of the SVF, secretome, hASCs, and AESCs on matrix protein secretion between healthy and OA chondrocytes, we conducted an ELISA assay.

COL10A1 was secreted in comparable amounts between all groups. Co-cultured OA–SVF and OA–AESCs induced COL1A1 and COMP biosynthesis.

Collagen type II was secreted in comparable amounts between all groups except in AESC-treated chondrocytes (Figure 8).

## 4. Discussion

Osteoarthritis is one of the main causes of pain and disability. This condition leads to progressive cartilage deterioration and a loss of function of the whole joint. Recent data shows how this condition affects more than 300 million people, representing a relevant health problem for patients with a condition of severe disability.

Despite its prevalence and importance, the molecular mechanisms underlying OA are still quite unclear. This lack of knowledge reflects the absence of proper and efficient treatment. Consequently, there is a consensus inside the scientific community to focus on the molecular mechanisms regulating OA etiology and development, with the purpose of highlighting key up- or down-regulated pathways that may dictate and control cartilage homeostasis and its behavior upon tissue injury.

The results of several preclinical and clinical studies have shown that, compared with traditional treatments, cell therapy can promote the formation of typical hyaline cartilage, with a positive score also in symptom control.

Mesenchymal stem cells (MSCs) are multipotent cells with the ability to self-renew and differentiate in cells of the mesodermal lineage (adipocytes, chondrocytes, and osteocytes). MSCs have been reported to have anti-inflammatory, anti-apoptotic, and anti-fibrotic properties. For these characteristics, they are of great interest in tissue engineering and in regenerative medicine. Many clinical studies performed in recent years show how mesenchymal stem cells, or SVFs, had a great impact on the quality of life of patients with osteoarthritis.

These studies show how cellular treatments have been successful in the management of OA [18,19]. Data in the literature show how both clinical trials using mesenchymal stem cells and SVFs have been successful in the management of osteoarthritis symptoms such as pain and stiffness. However, it is interesting to note that in the isolation of mesenchymal stem cells, it takes a long time for both isolation from tissues and the in vitro expansion to obtain adequate numbers for the treatment of the joints. The use of SVF for joint treatment does not require cell expansion or culture, and thanks to the simplicity of the protocol, it can be isolated at the bedside of the patient avoiding in vitro steps [19,20].

In addition, scientific research has shown how MSC’s secretome has an important role in regenerative medicine. The secretomes are all factors that MSCs release and consist of two components: extracellular vesicles (EVs) and soluble factors. Whole soluble factors, such as cytokines, chemokines, growth factors, and extracellular vesicles (EVs), secreted by MSCs can reduce the damage and improve the regenerative potential of the target tissues [21,22,23].

The use of secretomes is a new possible therapeutic approach in regenerative medicine because it preserves the same therapeutic properties of the original cell line but is free of the tumorigenic effect that you could have using cell lines [23].

In this article, we explored the therapeutic effects of mesenchymal stem cells from adipose tissue (hASCs), human amniotic epithelial stem cells (AESCs), SVFs, and the secretome derived from hASCs on cartilage regeneration.

Several studies show how mesenchymal adipose stem cells show a reparative and regenerative role for cartilage through paracrine signaling. Experimental evidence shows that these cells induce angiogenesis both in vitro and in vivo and regulate cell proliferation thanks to the variety of cytokines, chemokines, and growth factors secreted. Moreover, adipose tissue contains a greater number of stem cells compared to the same volume of other tissues, and the cells can harvested with a minimally invasive surgical procedure [24,25,26].

Several research lines are focusing on pluripotent stem cells as potential candidates for therapeutic use due to their higher self-renewal capabilities and proliferation rate with respect to MSCs.

Embryonic stem cells (ESCs) and induced pluripotent stem cells (iPSCs) have the risk of teratoma formation and immune rejection. For this reason, this research is concentrating on hASCs and human amniotic epithelial stem cells (AESCs) [27]. Currently, there are no in vitro studies that have investigated the role of AESCs in articular cartilage regeneration; however, there are several studies that reported that the amniotic membrane promotes tissue regeneration via several mechanisms: immuno-modulation and anti- inflammation, anti-fibrosis, pro- and anti-angiogenic properties, and ECM deposition [28,29].

Our analysis of the overall biological outcome of diseased chondrocytes through qPCR and ELISA assays shows different outcomes. The qRT-PCR analysis (shown in Table 2) of healthy chondrocyte markers shows an increase in *sox9*, *col2a1*, and *acan* at 7 and 21 dps in OA+ hASCs co-cultured while OA + AESCs shows an increase in *acan* and *col2a1* only at 7 dps. Moreover, the hypertrophic markers *runx2* and *col10a1* are downregulated in OA + hASCs and upregulated in OA + AESCs at 7 and 21 dps.

These data show that the treatment of OA chondrocytes with AESC induces significant changes in the qualitative composition of the ECM but improves hypertrophic and fibrotic conditions, so they do not appear to be a good candidate for the treatment of OA. Instead, treatment with hASCs promotes *sox9* and *acan* expression and represses *runx2*, *col10a1*, and *col1a1*, showing the ability of hASCs in high-quality cartilage regeneration.

It is interesting to note the expression of the fibrotic markers *mmp1* and *mmp13* are upregulated in OA + AESCs both at 7 and 21 dps, while OA + hASCs increased only *mmp13* expression at 7 dps.

In this work, we also analyzed the experimental data of co-cultures between OA chondrocytes + SVF and between OA chondrocytes + a secretome. There were no statistical differences in the expression of the healthy cartilage markers (*sox9*, *acan*, and *col2a1*) in the presence of the hASCs, SVF, or secretome. Only at 21 dps with SVF was there a significant increase in *col1a1* expression.

This data demonstrates that only the secretome was sufficient to trigger the regenerative effects in cartilage damage. Our results confirmed that paracrine signaling might be the main route of cell communication in wound healing and tissue regeneration.

The controversial issue about SVF is that it can also induce a fibrotic condition. Indeed, our results show a significant increase in *col1a1* at 7 and 21 dps and *runx2* at 21 dps in OA + SVF.

To evaluate the ECM proteins and cartilage-specific proteins between healthy and OA chondrocytes, we performed a quantitative ELISA analysis at 21 dps. The proteins evaluated were COL1A1, COL2A1, COL10A1, and COMP. Our ELISA experiments match the literature data and confirm an increase in OA chondrocyte proliferation and a reduction in the matrix biosynthesis process. The number of total proteins increased in OA chondrocytes compared to healthy ones, but there was a decrease in matrix proteins.

We also conducted an ELISA assay on matrix protein secretion between healthy and OA chondrocytes to investigate the role of the SVF, secretome, hASCs, and AESCs.

COL10A1 was secreted in comparable amounts between all groups. Co-culture OA + SVF and OA + AESCs induced COL1A1 and COMP biosynthesis.

Collagen type II is present in comparable amounts between all groups except in OA + AESCs. Protein evaluation by ELISA partially confirmed the results of gene expression; the discrepancies that emerged are probably due to molecular pathways that interact in the transcription–translation mechanism.

## 5. Conclusions

In our data, the application of the AESCs induced a quick and sharp gene expression of healthy but also fibrotic and hypertrophic markers. These results suggest that the AESCs could induce significant molecular changes in the diseased chondrocytes but also cause an upregulation of *col1a1*, which induces cartilage erosion and whole joint disruption.

The results of the co-cultures with SVF and AESC are similar; indeed, the application both of the AESCs and SVF induced a quick and sharp gene expression of healthy but also fibrotic and hypertrophic markers. So, this type of approach is not indicated for OA treatment.

The gene and protein expression analysis shows that the treatment with hASCs and a secretome could be the best biological approach for cartilage regenerative medicine. The co-culture OA-hASCs and a secretome displayed the same results for gene and protein expression. These data demonstrate that only the secretome was sufficient to trigger the regenerative effects in cartilage damage. Our results confirmed that paracrine signaling might be the main route of cell communication in wound healing and tissue regeneration.

In summary, we can confirm that the therapeutic approach with hASCs and their secretome is of great interest as a possible therapy to relieve or delay the symptoms of osteoarthritis. However, there are still many aspects to clarify, and many in vitro and in vivo studies are still necessary.

## Figures and Tables

**Figure 1 biomolecules-14-01068-f001:**
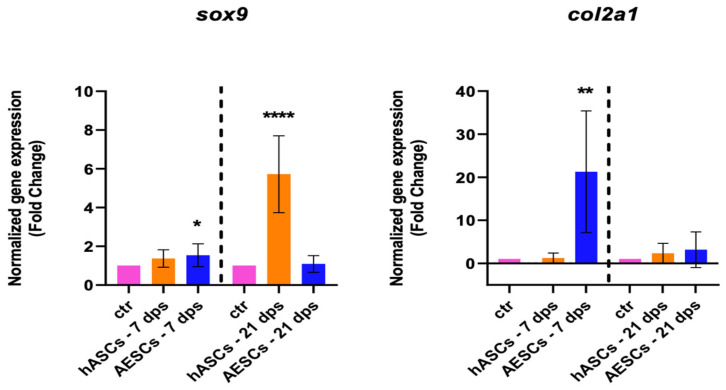
Healthy chondrocytes marker expression results of OA chondrocytes co-cultured with either the multipotent (hASCs) or pluripotent (AESCs) stem cells for 7 and 21 dps. The raw data were tested for normality and then analyzed with a one-way ANOVA followed by Tukey’s and Dunnett’s tests for multiple comparisons. The differences were considered statistically significant when: * *p* < 0.05; ** *p* < 0.005; *** *p* < 0.0005; **** *p* < 0.001.

**Figure 2 biomolecules-14-01068-f002:**
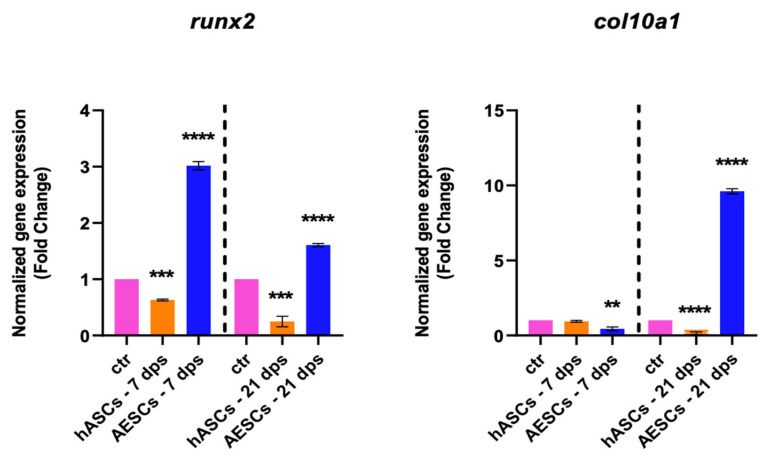
Hypertrophic marker expression results of OA chondrocytes co-cultured with either the multipotent (hASCs) or pluripotent (AESCs) stem cells for 7 and 21 dps. The raw data were tested for normality and then analyzed with a one-way ANOVA followed by Tukey’s and Dunnett’s tests for multiple comparisons. The differences were considered statistically significant when: ** *p* < 0.005; *** *p* < 0.0005; **** *p* < 0.001.

**Figure 3 biomolecules-14-01068-f003:**
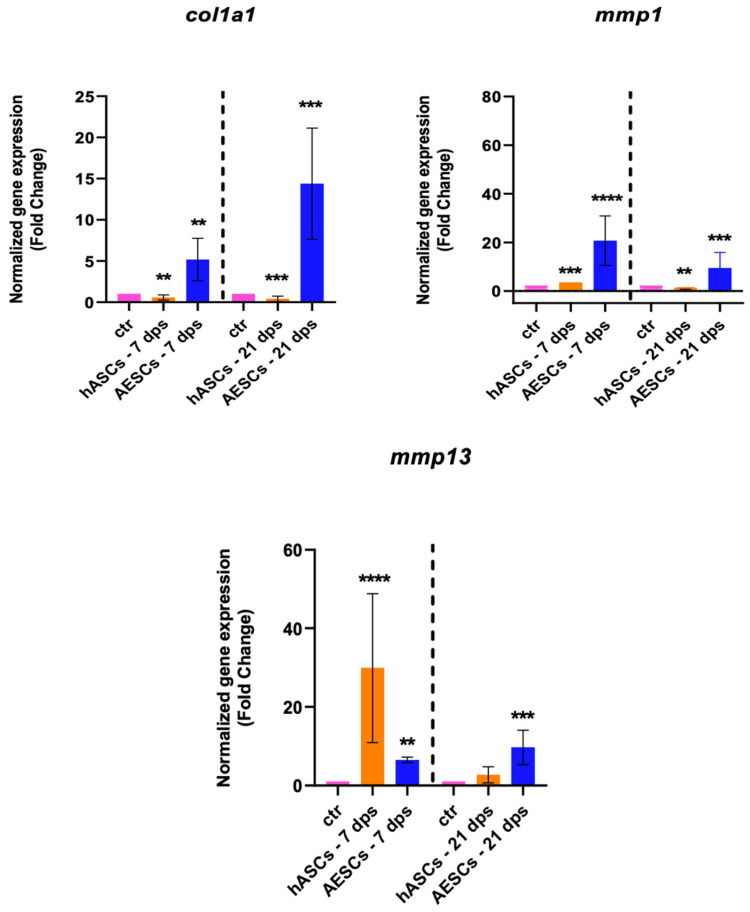
Fibrotic marker expression results of OA chondrocytes co-cultured with either the multipotent (hASCs) or pluripotent (AESCs) stem cells for 7 and 21 dps. The raw data were tested for normality and then analyzed with a one-way ANOVA followed by Tukey’s and Dunnett’s tests for multiple comparisons. The differences were considered statistically significant when: ** *p* < 0.005; *** *p* < 0.0005; **** *p* < 0.001.

**Figure 4 biomolecules-14-01068-f004:**
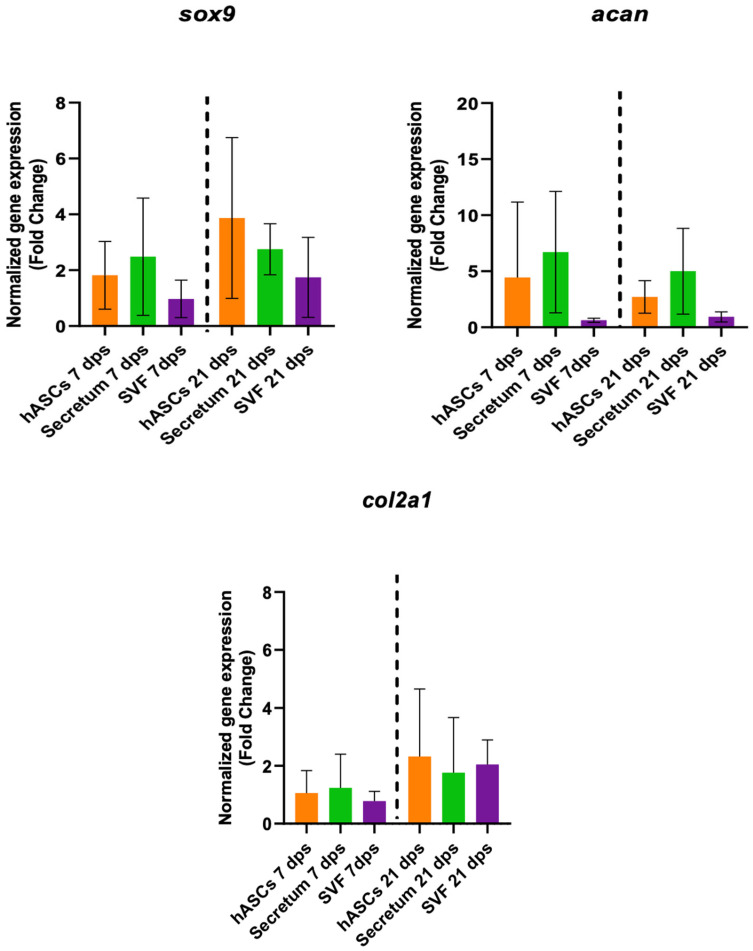
Gene expression results of OA chondrocytes cultured with the multipotent stem cells (hASCs), SVF, and their derivatives (secretum) for 7 and 21 dps. The bar plots show healthy cartilage markers. The raw data were tested for normality and then analyzed with a one-way ANOVA followed by Dunnett’s test for multiple comparisons. The differences were considered statistically significant when *p* < 0.05.

**Figure 5 biomolecules-14-01068-f005:**
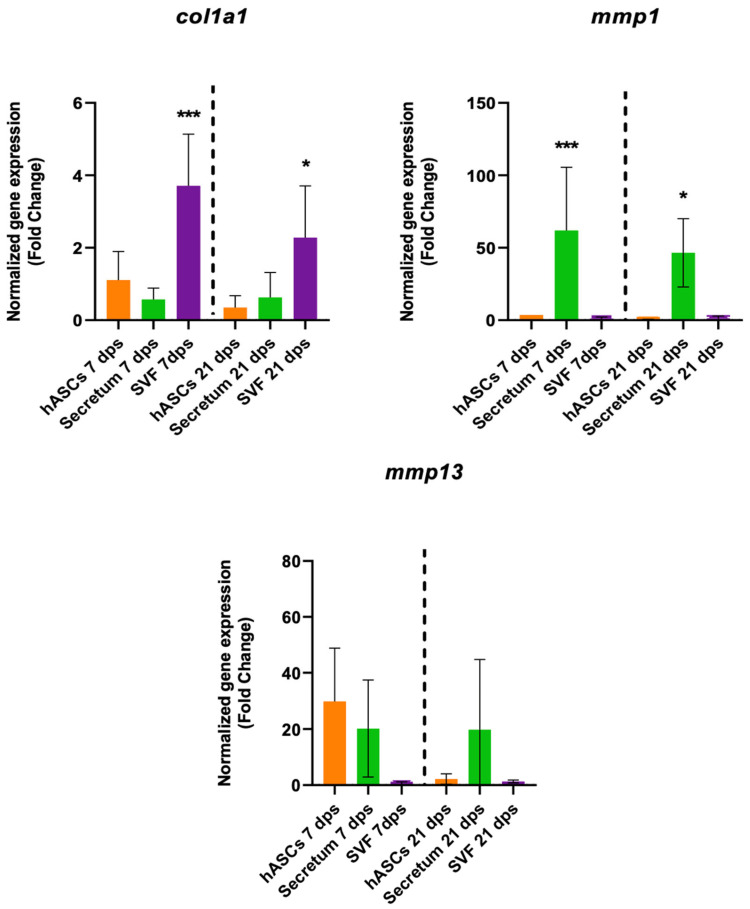
Gene expression results of OA chondrocytes cultured with the multipotent stem cells (hASCs and SVF) and their derivatives (secretum) for 7 and 21 dps. The bar plots show the fibrotic markers. The raw data were tested for normality and then analyzed with a one-way ANOVA followed by Dunnett’s test for multiple comparisons. The differences were considered statistically significant when: * *p* < 0.05; *** *p* < 0.0005.

**Figure 6 biomolecules-14-01068-f006:**
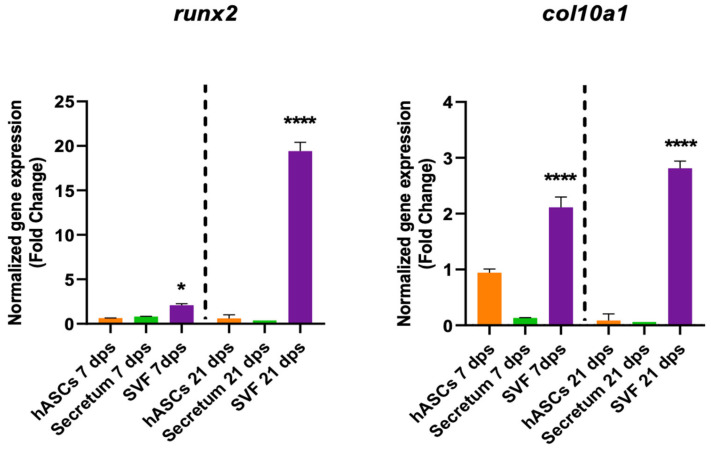
Gene expression results of OA chondrocytes cultured with the multipotent stem cells (hASCs and SVF) and their derivatives (secretum) for 7 and 21 dps. The bar plots show the hypertrophic (degenerative) markers. The raw data were tested for normality and then analyzed with a one-way ANOVA followed by Dunnett’s test for multiple comparisons. The differences were considered statistically significant when: * *p* < 0.05; **** *p* < 0.001.

**Figure 7 biomolecules-14-01068-f007:**
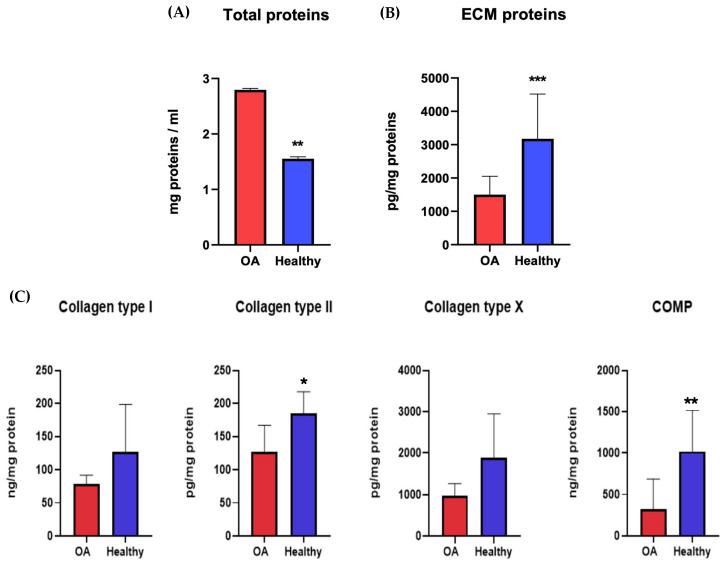
Total protein content (**A**) and ECM protein (**B**,**C**) concentrations of healthy and diseased (OA) chondrocytes. The raw data were tested for normality and further analyzed with an unpaired *t*-test. The differences were considered statistically significant when: * *p* < 0.05; ** *p* < 0.005; *** *p* < 0.0005.

**Figure 8 biomolecules-14-01068-f008:**
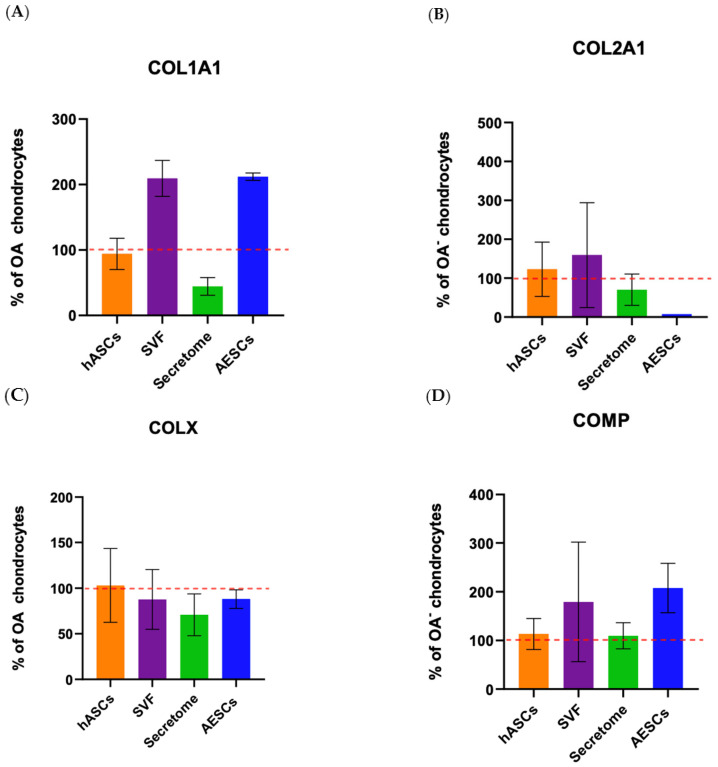
Protein expression of OA chondrocytes treated with the pluripotent (AESCs) and multipotent stem cells (hASCs and SVF) and their derivatives (secretum) at 21 dps. The relative protein concentrations in the bar plots are expressed as % over the expression of healthy chondrocytes. The raw data were tested for normality and then analyzed with a one-way ANOVA followed by Tukey’s and Dunnett’s tests for multiple comparisons. The differences were considered statistically significant when *p* < 0.05. In (**A**), the expression of collagen type I; in (**B**), the expression of collagen type II; in (**C**), the expression of collagen X; and in (**D**), the expression of COMP. The reference sample is healthy chondrocytes (dashed red line).

**Table 1 biomolecules-14-01068-t001:** Primers used in real-time PCR experiments.

Gene	Target Transcript	Amplicon Length
*COL1A1*	NM_000088	127 bp
*COL2A1*	NM_001844	121 bp
*COL10A1*	NM_000493	150 bp
*ACAN*	NM_013227	85 bp
*SOX9*	NM_000346	118 bp
*RUNX-2*	NM_004348	102 bp
*MMP1*	NM_002421	102 bp
*MMP13*	NM_002427	130 bp
*β2M*	NM_004048	98 bp

**Table 2 biomolecules-14-01068-t002:** Summary of qRT-PCR data.

Healthy Chondrocytes Markers			
OA + AESC	*Sox9* -	*Col2a1* ↑	*Acan* ↑
OA + hASC	*Sox9* ↑	*Col2a1* -	*Acan* ↑
OA + SVF	*Sox9* ↑	*Col2a1* -	*Acan* ↑
OA + secretome	*Sox9* ↑	*Col2a1* -	*Acan* ↑
**Hypertrophic chondrocytes markers**			
OA + AESC	*Runx2* ↑	*Col10a1* ↑	
OA + hASC	*Runx2* ↓	*Col10a1* ↓	
OA + SVF	*Runx2* ↑	*Col10a1* ↑	
OA + secretome	*Runx2* ↓	*Col10a1* ↓	
**Fibrotic chondrocytes markers**			
OA + AESC	*Col1a1* ↑	*Mmp1* ↑	*Mmp13* ↑
OA + hASC	*Col1a1* ↓	*Mmp1* ↓	*Mmp13* ↑
OA + SVF	*Col1a1* ↑	*Mmp1* ↓	*Mmp13* ↑
OA + secretome	*Col1a1* ↓	*Mmp1* ↑	*Mmp13* ↑

Legend: up-regulation gene expression (↑), down-regulation gene expression (↓), no changes in gene expression (-).

## Data Availability

Data are contained within the article.

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
