# Peer review of "Mesenchymal Stem Cells and Secretome as a New Possible Approach to Treat Cartilage Damage: An In Vitro Study"

_biomolecules, 2024, doi:10.3390/biom14091068_

Round 1

Reviewer 1 Report

Comments and Suggestions for Authors

The article “Therapeutical strategies for cartilage repair: an in vitro study” investigated the actual issue of clarifying the regenerative effects of multipotent adipose derived stem cells and the pluripotent amniotic epithelial stem cells on arthrosis chondrocytes performing co-culture experiments.

The most important health problems in our society are damage and degeneration of peripheral or central (vertebral) joints, which are associated with the limited ability of the tissue to regenerate. According to statistics, knee osteoarthritis is one of the main factors of disability with pain and stiffness, resulting in severe functional limitations. Osteoarthritis is known to be one of the most common musculoskeletal diseases and poses a huge burden on health and social care systems and countries’ economies worldwide.

In fact, there are some points to be addressed.

1. The title should describe the essence of the study in more detail to make it easier for the reader to understand.

2. In the ‘Abstract’ section, you should focus on the objects of research, research methods, and results and conclusions.

3. When separating integer and fractional parts of numbers, "," must be replaced with "." (for example, lines 125 and 130).

4. It is necessary to clarify which passages of OA chondrocytes, AESCs and hASCs were used in the study.

5. Please, specify the total protein measurement method.

6. It is also necessary to add the 'Conclusions' section.

7. The methods for obtaining SVF and secretome should be described.

8. An explanation for the observed differences in the effect of cells and their secretome on OA chondrocytes should be offered.

9. A list of gene primers should be added.

10. Please, specify how many micromasses were used to quantify each parameter.

11. The results should be discussed in more detail.

12. In terms of the English language, unfortunately, numerous inaccuracies have been spotted, such as grammar issues, word order, complete sentences, etc. For example, lines 44-45, 84-85, 86, 164, 256, 287-289, 405, 419, 428, 468, 474.                                                                                                                                                                                                                                                              

Comments on the Quality of English Language

In terms of the English language, unfortunately, numerous inaccuracies have been spotted, such as grammar issues, word order, complete sentences, etc. For example, lines 44-45, 84-85, 86, 164, 256, 287-289, 405, 419, 428, 468, 474.          

Author Response

We thank the reviewers for their helpful advice, we have corrected the article following your recommendation.

Reviewer 1

1.The title should describe the essence of the study in more detail to make it easier for the reader to understand.

We change the title as suggest by the reviewer.

2.In the ‘Abstract’ section, you should focus on the objects of research, research methods, and results and conclusions.

Thanks for the tip, we changed it.

3. When separating integer and fractional parts of numbers, "," must be replaced with "." (for example, lines 125 and 130).

We do it.

4.It is necessary to clarify which passages of OA chondrocytes, AESCs and hASCs were used in the study.

We do it in line 232.

5. Please, specify the total protein measurement method.

We do it in line from 447-449.

6.It is also necessary to add the 'Conclusions' section.

Thanks for the tip, we add the conclusions section.

7. The methods for obtaining SVF and secretome should be described.

We add the methods for obtaining SVF at line 206-210. While the secretome is purchased (line 224).

8.An explanation for the observed differences in the effect of cells and their secretome on OA chondrocytes should be offered.

Our experimental show a same effect between hASC and secretome (present in the discussion)

9.A list of gene primers should be added.

Thanks for the tip, we add a table with the list of primer used in material and methods

10.Please, specify how many micromasses were used to quantify each parameter.

Thanks for the tip, we add at line 236-237

11.The results should be discussed in more detail.

Thanks for the tip, we do it in discussion sections.

12.In terms of the English language, unfortunately, numerous inaccuracies have been spotted, such as grammar issues, word order, complete sentences, etc. For example, lines 44-45, 84-85, 86, 164, 256, 287-289, 405, 419, 428, 468, 474.

Thanks for the tip, we have revised the English and corrected the various grammatical errors.   

Reviewer 2 Report

Comments and Suggestions for Authors

The research manuscript titled, “Therapeutical strategies for cartilage repair: an in vitro study”, tries to explain the effects of various therapeutical options such as different stem cells, secretome etc., during the chondrocyte culture. The authors have carried out extensive studies by analyzing expression of various gene and protein expressions. The results would be very useful for researchers working on chondrocyte regeneration and repair and also for the clinicians. The manuscript will require minor revisions before publications, as follows:

1.      Line 13 and line 50, “…often common in occidental countries..”, Try to avoid specific regions. If specified, then reference for the data source is required.

2.      Line 16, check grammar.

3.      Line 24, line 99 and few other places, the authors have used the term ‘liosecretome’. The authors are advised to check the accuracy of this term and quote references for the term.

4.      Include the result of the study briefly in the abstract.

5.      The authors are advised to use uniform decimal format. Dot (0.25) instead of comma (0,25) wherever applicable.

6.      In section “isolation of human adipose-derived stem cells”, include the approval numbers.

7.      In materials and methods section, include the protocol about using SVF.

8.      Isolation protocols of AESCs have to be added.

9.      Line 348, Spell check.

10.   In discussion, the authors could compile and tabulate their results by listing the components (AESC, SVF, etc) and indicating the various parameters that were increased or decreased. This would help the readers to easily understand the results.

11.   Line 470, check sentence.

Author Response

We thank the reviewers for their helpful advice, we have corrected the article following your recommendation.

Reviewer 2

1.Line 13 and line 50, “…often common in occidental countries..”, Try to avoid specific regions. If specified, then reference for the data source is required.

Thanks for the tip, we do it.

2.Line 16, check grammar.

Thanks for the tip, we do it.

3.Line 24, line 99 and few other places, the authors have used the term ‘liosecretome’. The authors are advised to check the accuracy of this term and quote references for the term.

Thanks for the tip, we changed the term with one more appropriated.

4.Include the result of the study briefly in the abstract.

Thanks for the tip, we changed it.

5.The authors are advised to use uniform decimal format. Dot (0.25) instead of comma (0,25) wherever applicable.

Thanks for the tip, we changed it.

6.In section “isolation of human adipose-derived stem cells”, include the approval numbers.

Thanks for the tip, we add the Ethics Committee number in the section: “isolation of hASC” at line 197-198.

7. In materials and methods section, include the protocol about using SVF.

Thanks for the tip, we add the protocol in line 206-2010.

8. Isolation protocols of AESCs have to be added.

Thanks for the tip, unfortunately we do not have the protocol for the isolation of AESCs, because in our laboratory it is not possible to isolate these cells. The cells used in these experiments were kindly donated by obstetrics unity of Poliambulanza Foundation, Brescia (line 173-175)

9. Line 348, Spell check.

Thanks for the tip, we correct it.

10.In discussion, the authors could compile and tabulate their results by listing the components (AESC, SVF, etc) and indicating the various parameters that were increased or decreased. This would help the readers to easily understand the results.

Thanks for the tip, we add a table with summary the results.   

11.Line 470, check sentence.

Thanks for the tip, we correct it.

Round 2

Reviewer 1 Report

Comments and Suggestions for Authors

I thank the authors of the article “Therapeutical strategies for cartilage repair: an in vitro study” for fixing most of the point highlighted. Unfortunately, the English language of the article remains an obstacle that deprives the reader of the full understanding of the issues covered.

I strongly recommend the authors to submit the article to a professional linguist for the article’s language thorough revision.

Comments on the Quality of English Language

Unfortunately, the English language of the article remains an obstacle that deprives the reader of the full understanding of the issues covered.

I strongly recommend the authors to submit the article to a professional linguist for the article’s language thorough revision.